# Unveiling parents' lived experience with preterm infant care and support in neonatal care units of public hospitals in Nepal: A phenomenological inquiry

Tumla Shrestha[1]*, Archana Pandey Bista[1], Sarala Joshi Pradhan[2],
Sangita Pudasainee-Kapri[3], Madhusudan Subedi[4]

**1** Maharajgunj Nursing Campus, Institute of Medicine, Tribhuvan University, Kathmandu, Bagmati, Nepal,
**2** Om Health Campus, Purbanchal University, Kathmandu, Bagmati, Nepal, **3** The Rutgers State University
of New Jersey, School of Nursing-Camden, New Brunswick, New Jersey, United States of America,
**4** Patan Academy of Health Science, Lalitpur, Bagmati, Nepal

* tumlashrestha@gmail.com

## Abstract

### Background

Preterm infants (PTIs) require hospitalization in different levels of neonatal care units (NCUs) for their survival and developmental needs. The quality of care provided at NCUs significantly influences infant outcomes and parents' experiences. Parents' experience of received support and care of PTIs is one of the indicators for determining the quality of care at NCUs. The study aims to investigate parents' perspectives on the PTIs care and support received from nurses in NCUs of Nepal.

### Methods

A descriptive phenomenological study was conducted within the NCUs of three public tertiary hospitals in Kathmandu, Nepal. In-depth interviews were conducted among 25 purposively selected parents, (both mothers and fathers) of low-birthweight PTIs admitted to the NCUs. Data was collected from November 2019 to February 2020. The data were meticulously analyzed using the Colaizzi method.

### Findings

The exploration of parents' experiences identified three main theme areas: (1) Care and support, (2) Initial involvement in PTI care, and (3) Outcome of care involvement. Parents appreciated competent and affectionate PTI care as well as informational support. However, they had varied experiences with communication, emotional support, and opportunities for infant-parent attachment. Guidance and support for PTI care from nurses and peer-parents proved instrumental in mitigating uncertainties related to initial care learning and involvement in PTI care. Parents' involvement in hands on care of their PTIs boosted infant-parent attachment, empowered for care giving, and provided emotional solace.

**Data availability statement:** Most of the relevant data are within the paper and remaining data are attached as Supporting information: "Data Analysis Matrix, Care and Support.

**Funding:** The author(s) received no specific funding for this work.

**Competing interests:** The authors have declared that no competing interests exist.

**Abbreviations:** EBM, Expressed breast milk; IDI, In-depth interview; KMC, Kangaroo mother care; NCU, Neonatal care unit; NICU, Neonatal intensive care unit; PTI, Preterm infant; SNCU, Sick newborn care unit

## Conclusion

Findings indicate that parents have positive experience with PTI care provided by nurses and their involvement in hands-on care of their PTIs. However, there are gaps in support expectations of parents including communication, emotional support, and care guidance. Findings have important implications for nurses, pediatricians, and policymakers for the enhancement of neonatal care practice by incorporating parental support and parents' involvement in hands on care of PTI across NCUs in Nepal.

## Introduction

Preterm birth (PTB) is a live birth that occurs before 37 completed weeks of gestation. Based on infants' gestational age, it is further classified into four categories including extreme (before 28 weeks); very (28–31 weeks), moderate (32–33 weeks), and late (34–37 weeks) preterm birth [1]. Among them, late PTB (34–37 weeks) are more common. According to Lancet Global Health, there were an estimated 13.4 million preterm births in 2020, constituting 9·9% of all births worldwide. This reflects a slight decrease from 13.8 million recorded in 2010, accounting for 9·8% of total births [2]. South Asia has the highest preterm birth rate (13.2%) [3]. In Nepal, the incidence of preterm birth is reported between 9.3% to 14% across studies [4,5].

Prior study found that Preterm infants (PTIs) have substantially higher risks for neonatal and post-neonatal mortality, morbidity, and disability [6–8]. The risks are inversely proportionate to gestational age and PTIs' weight at birth. The mortality and morbidity risk along with the length of hospital stay decreases significantly with each additional completed weeks of gestation among PTB infants. Globally, prematurity and its related complications were the leading cause of neonatal and under-five mortality [9].

PTIs require special care in various tiers of neonatal care units (NCUS) for survival and proper development with admission ranging from a few days to several weeks to months [10,11]. The quality of neonatal care provided determines the survival and outcome of PTIs. The care offered at NCUs also affects parental satisfaction. Thus, parents' experience during PTI's hospitalization is one of major indicators for determining quality of provided care [12].

However, hospitalization of PTIs in NCUs is stressful experience for parents. Their stress is related to their PTI's medical condition and potential outcomes as well as other challenges associated with the longer hospitalization. Furthermore, parents especially mothers experience higher stress related to infant-parents separation, and altered parental roles [13–17]. Those parents often experience a sense of powerlessness, feeling uninformed, and intimidated in the highly specialized care environment of NCUs. Parental stress as well as disrupted parent–infant interaction/bonding may adversely affect parental engagement in infant care, care enabling and infant outcomes [18]. Therefore, parents especially mothers should be supported by NCU staff by respecting their views and needs, answering parents' questions, providing adequate information, opportunities for attachment and care engagement in friendly environment [18,19]. As a frontline cadre in NCUs, nurses are in key positions to support parents and assist them with coping [20,21].

Existing literature highlighted that communication among parents of PTIs and healthcare providers are the core of parents' experience [22,23]. In particular, qualitative studies and systematic review indicate that receiving adequate information and timely communication about the PTI's condition and care-giving activities, trusting relationships with NCU staff, opportunities for parents to be closer to their infants, and parental involvement in PTIs' care

with hands on experiences are very important for effective coping, PTI care enabling, and care satisfaction [19,23–27]. In addition, parents need a parents friendly environment, emotional support encouragement and, mutual respect for care engagement [28]. Parent-to-parent support (i.e., support and experience sharing from parents of other PTIs admitted in NCUs) is also effective for emotional comfort and for care engagement.

Parents expect support and an enabling environment from health professionals along with hands-on care experience to develop parenting skills in providing basic care to PTI such as bathing, feeding, changing diapers and massaging (care involvement) [29,30]. Nurses have a key role to create such environment for parents engagement in caring PTIs at NCUs by acknowledging the parents as the primary caregiver and being aware of the benefits of parents' involvement in PTI care to both infants and families [25]. Parents valued proper guidance and demonstration of skills related to PTI care like KMC followed by support to undertake skills by themselves [23,27,31]. Parents like to learn and actively engage in their PTIs care and ultimately develop self-efficacy and confidence in providing PTI care as this is useful for transitioning from the NCUs to the mother-baby unit and the hospital-home transition. Existing literature showed that parental involvement in PTI care enhances psychological wellbeing among parents, infant-parent attachment, and PTI care confidence [32,33].

Parents' expectations of attributes like friendly, kind, caring, empathetic, and supportive behavior among NCUs staff are core to nursing and caring sciences. However, parents in NCUs do not always experience such behaviors due to increased staff workload and unit policies regarding parents' roles and restrictive vising hours in NCUs etc [26,32]. Previous studies in various context [34,35] indicated that parental support from nurses, exhibited through active listening to parents' concerns, providing individualized support, and encouraging participation in their PTI's care promote a positive experience for parents [34,35]. However, other studies reported inadequate information sharing, improper communication and emotional support provided by nurses [36–40]. Prior studies also reported some mixed or inconsistent findings such as adequate and/or inadequate opportunity for attachment with PTIs and support for involvement in care of their PTIs [14,15,41]. Limited research to date has examined parents' (including both mothers and fathers) perspectives regarding PTI care and parental support provided in NCUs in low-middle-income countries specifically in the context of Nepal. Examining those concepts is vital since proper parental engagement and support in PTIs care enhance infant outcomes and parental readiness in caring at-risk low birth weight PTIs infants. Therefore, this study was conducted to explore the experience of mothers and fathers of PTIs receiving care and support in NCUs during admission of their PTIs.

## Methods

### Design and settings

The descriptive phenomenological study was conducted in NCUs of three public tertiary hospitals in Kathmandu, Nepal including Tribhuvan University Teaching Hospital (TUTH), Paropakar Maternity and Women's Hospital (PMWH), and Kanti Children's Hospital (KCH). Neonatal care units included both sick newborn care units (SNCU) or level II neonatal care units and neonatal intensive care units (NICU) or level III neonatal care units. TUTH and KCH served as central referral hospitals receiving PTIs in their NCUs from various hospitals of the capital city and across the country. PMWH is a government hospital that conducts approximately 70 to 90 births per day with a significant number of newborns requiring admission to the NCUs. TUTH has 19 neonatal beds with 9 in NICU and 10 in SNCU. PTIs were transferred to mother-baby units for a few days before being discharged home. PMWH has 40 neonatal beds, including 10 in the NICU, 26 in the SNCU, and 4 in the KMC unit. PTIs of weight <1500 grams

were transferred to the KMC unit for continuous KMC till satisfactory weight gain for discharge. KCH is the oldest public tertiary-level children's hospital in Nepal. The NCU of KCH had 12 beds in NICU (functioning) and 15 beds in SNCU at study time. SNCU was named as neonatal intermediate care unit (NIMCU). In KCH, mothers were involved in hands on care of their PTIs such as breast milk feeding, KMC, and hygiene care in SNCU before discharge. Usually, PTIs who were improved from critical condition were transitioned to a SNCU. SNCU also received admission of sick PTIs directly from outpatient or Emergency departments. PTIs recovering were transferred from SNCU to mother-baby unit/KMC unit before discharge.

## Participants

The study participants were mothers and fathers of PTIs (born before 37 weeks of gestational age) with low birth weight (birth weight less than 2500 grams) who were admitted into the NCUs of one of three hospitals in Nepal. To obtain a wider range of experiences, parents from diverse socio-demographic backgrounds (e.g., age group, educational status and occupational status) were included. Likewise, mothers with varied obstetric backgrounds (e.g., operative and vaginal deliveries, including home deliveries, as well parents of PTIs with various degrees of low birth weight and gestational age were included. Both parents of PTIs residing in the capital city as well as referred from countryside for management of infants and mothers were also included in the study (Tables 1 and 2).

Parents whose PTIs had no congenital malformations, metabolic or genetic disorders and stayed in a NCUs for four days or more were recruited in the study. Participants were recruited after their PTIs had recovered and were scheduled for discharge from the NCUs to mother-baby units. Eligible parents who were willing to share their experiences were purposively selected for the interview.

Parents who provided consent for possible repeated interviews (to get in-depth data from the same participant) were recruited until sample saturation was reached. Both parents were included in the study. More mothers were recruited for interviews as they had more parenting role and were more easily available than fathers during data collection. Either the mother or the father of a PTI was included in the study. The final sample for this study was 25 participants consisting of 20 mothers and 5 fathers of PTIs from three settings between 10 November 2019 to 26 February 2020.

## Ethical considerations

The study was approved by the Nepal Health Research Council (Ref. No. 2804/2019). and administrative approval for data collection was obtained from each of the hospitals. Written informed consent was obtained from each participant as well as audio-recording of the interviews. Audio records were kept in a password-protected laptop and transcription documents were kept confidential. Precaution was taken to present personal information while presenting the study findings.

## Instrument and data collection methods

A semi-structured interview guide was developed to interview parents of PTIs that aligned with the purpose of the study and available literature. The interview guide included directions for conducting in-depth interviews, socio-demographic questionnaires, and open-ended questions. The instrument was first made in English language and then translated into simple Nepali language since that is the native language of the participants.

For the enhancement of the trustworthiness of the instrument, relevant literature was reviewed, and subject matter experts and a qualitative research expert were consulted. A

**Table 1. Descriptive Characteristics of the Study Participants.**

| Participant Number | Age (years) | Ethnic Group | Educational Status | Occupational Status | Residence Status | Parity | Family Type |
|---|---|---|---|---|---|---|---|
| **Mothers** | | | | | | | |
| 1 | 25 | Madhesi | Secondary | School teacher | Mother referred | Second | joint |
| 2 | 28 | Janajati | Secondary | Homemaker | Mother referred | Second | Nuclear |
| 3 | 40 | Chhetri | Master | Business | Mother referred | Second | Joint |
| 4 | 23 | Janajati | Secondary | Homemaker | Infant referred | Primi | Joint |
| 5* | 40 | Janajati | Can't read & write | Daily wage earner | Kathmandu | Second | Nuclear |
| 6 | 23 | Bramhan/ Chhetri | Secondary | Homemaker | Infant Referred | Primi | Joint |
| 7 | 25 | Janajati | Higher Secondary | Homemaker | Kathmandu | Primi | Joint |
| 8** | 20 | Janajati | Basic | Homemaker | Mother referred | Second | Joint |
| 9 | 23 | Madhesi | Higher Secondary | Service (School teacher) | Infant referred | Primi | Joint |
| 10 | 18 | Chhetri | Secondary | Business | Kathmandu | Second | Nuclear |
| 11 | 25 | Chaudhari | Higher Secondary | Service Holder | Mother referred | Primi | Nuclear |
| 12 | 28 | Janajati | Informal | Homemaker | Kathmandu | Primi | Nuclear |
| 13*** | 30 | Janajati | Basic | Business (tailor) | Kathmandu | Second | Nuclear |
| 14 | 30 | Janajati | Bachelor | Service | Kathmandu | Primi | Joint |
| 15 | 27 | Chhetri | Basic | Homemaker | Kathmandu | Second | Nuclear |
| 16 | 19 | Dalit | Secondary | Homemaker | Kathmandu | Primi | Joint |
| 17 | 28 | Dalit | Can't read & write | Homemaker | Kathmandu | Second | Joint |
| 18 | 21 | Janajati | Basic | Homemaker | Mother referred | Second | Nuclear |
| 19 | 26 | Janajati | Secondary | Business | Kathmandu | Primi | Nuclear |
| 20 | 20 | Dalit | Basic | homemaker | Kathmandu | Primi | Nuclear |
| **Fathers** | | | | | | | |
| 1 | 32 | Janajati | Higher Secondary | Army | Mother referred | Second | Nuclear |
| 2 | 40 | Chhetri | Bachelor | Business | Kathmandu | Second | Joint |
| 3 | 24 | Janajati | Secondary | Overseas migrant worker | Mother referred | Primi | Joint |
| 4 | 31 | Janajati | Secondary | Service | Kathmandu | Second | Nuclear |
| 5 | 22 | Dalit | Secondary | Painting | Mother referred | Primi | Joint |

**Note:** * No ANC Care, has previous infant also (newborn is second child); **only one ANC visit, ***Prior infant death (8 months of age).

pilot study was conducted with five mothers of PTIs admitted to NCU and minor changes in the interview guide were made based on feedback and suggestions from the pilot study participants.

The lived experience of the parents was explored through in-person, in-depth interviews (IDIs) conducted between November 2019 to February 2020. Eligible parents were identified from NCUs of study settings and approached when there was a plan to transfer PTIs from the NCUs to mother-baby unit (TUTH) or KMC unit (PMWH) or planned for discharge (KCH). The written informed consent was obtained after explaining the purpose and significance of the study along with their role in the study and the option for voluntary participation. Interviews were conducted after shifting the PTIs into mother-baby unit of TUTH, and/or KMC unit in PMWH and 1-2 days before discharge from SNCU in KCH. Interviews were conducted in a separate quiet room considering the privacy and comfort of the parents and suitable for recording. The first author conducted the IDIs using a semi-structured IDI guide in the colloquial Nepali language. The interviewer explained her position as a research scholar to participants and that there was no affiliation to the data collection settings. The interviewer has had qualitative research and data management training and had completed a prior

**Table 2. Characteristics of Preterm Infants.**

| PN | Gestational Age (weeks) | Birth Weight (grams) | Cause of Preterm Birth | Type of Delivery | Hospitalization Duration |
|----|-----|------|------|------|------|
| 1 | 28 | 1400 | Antepartum hemorrhage | Operative | 24 |
| 2 | 28 | 1140 | PIH | CS | 30 |
| 3 | 28 | 1000 | Antepartum hemorrhage | CS | 41 |
| 4* | 30 | 1470 | PROM | Normal | 20 |
| 5 | 30 | 1500 | PROM | Normal (home) | 21 |
| 6 | 32 | 1700 | PROM | Normal | 20 |
| 7 | 30 | 1345 | PIH, Eclampsia | CS | 13 |
| 8 | 28 | 1300 | Antepartum hemorrhage | Normal | 26 |
| 9* | 30 | 1300 | PROM | Normal | 32 |
| 10 | 33 | 1880 | PROM | CS | 10 |
| 11 | 32 | 1250 | PIH | Normal | 21 |
| 12 | 29 | 1200 | PROM | CS | |
| 13 | 34 | 1500 | PROM, fetal distress | CS | 10 |
| 14 | 31 | 1700 | Spontaneous | Normal | 14 |
| 15 | 30 | 1300 | PROM | CS | 15 |
| 16 | 29 | 1800 | Spontaneous | Normal | 22 |
| 17 | 34 | 1530 | Spontaneous | Normal | 20 |
| 18 | 30 | 1600 | Spontaneous | Normal | 18 |
| 19 | 33 | 1780 | PROM | Normal | 10 |
| 20 | 34 | 1600 | Spontaneous | Normal | 8 |
| 1 | 28 | 1250 | PIH | CS | 30 |
| 2 | 28 | 1200 | Spontaneous | CS | 35 |
| 3 | 30 | 1450 | PROM | Normal | 20 |
| 4 | 34 | 1500 | PROM, fetal distress | CS | 10 |
| 5 | 29 | 1700 | Spontaneous | Normal | 22 |

*Second time admission *in* NCUs, PROM: premature rupture *of* membrane, PIH: Pregnancy induced hypertension, CS: caesarean section

qualitative study [42]. The study was piloted among five mothers of PTIs and the findings were published [17]. The interviews were audio-recorded on digital recorders and any verbal and non-verbal expressions and special moments were noted by the interviewer. The duration of the IDI sessions ranged from 30-60 minutes for the first-time interviews. Each participant was interviewed for an additional two to three sessions until data saturation was achieved.

## Data analysis

Data collection and analysis were conducted simultaneously by the first author and periodically discussed with other authors. The audio recordings were listened to and transcribed in Nepali. Transcriptions were read several times, co-related with the field notes, then translated into the English language. The data obtained were analyzed using Colaizzi's method of data analysis.

During data analysis, significant statements were extracted from each transcript. Initial meanings were formulated and arranged into thematic clusters. Those data were then combined to form a distinctive construct of sub-themes and themes. Sub-themes and themes were checked against the data to validate the emerging patterns. The themes and sub-themes were thoroughly described relating to significant statements (quotes).

## Trustworthiness

To ensure the quality rigor in the study, Lincoln and Guba's (1985) criteria–credibility, dependability, transferability, confirmability and authenticity of the qualitative research were followed [43,44]. Credibility was ensured through expert consultation and recruiting participants with experience of having their PTIs in NCUs. Multiple IDIs were conducted, and field notes were kept. Some participants (M2, M3, M11, M16, F1) were invited to provide feedback on the findings to ensure accurate interpretation of their expression (member checking). Dependability was maintained by using the same interview guide in each IDI and preparing transcripts within a few days of interviews. For confirmability, the co-authors reviewed the data analysis process and outcomes. For transferability, the research process and the findings were thoroughly presented, with quotes used to illustrate findings.

Authenticity is the extent to which researchers fairly and faithfully show a range of realities. To maintain the authenticity criterion of trustworthiness informed consent was obtained before data collection with detail explanation of the study and its purpose. Their queries and concerns were answered honestly before data collection. Likewise, trust and honest relationships were maintained with participants throughout the study. The informed consent was reaffirmed from time to time as new contingencies were faced during interviews. Participants were well informed about their voluntary participation. Participants of different socio-economic and obstetric backgrounds were included in the study to obtain data from diverse perspectives. Reflection of participants' views and expressions was done during data collection and member checking was done after data analysis. Furthermore, findings were presented including verbatim quotes from the participants and peer debriefing was also done with other authors.

## Findings

**Characteristics of participants and their PTIs.** Participants' age ranged from 18 to 40 years. More than half of the participants (n = 13) were from Janajati (indigenous) groups. They had diverse educational backgrounds including illiterate to master's degree and 14 had secondary-level education. Furthermore, 11 were homemakers (unpaid household work), 13 belonged to joint families and were first-time parents. Fifteen participants resided in the capital city Kathmandu as either temporary or permanent residents and 10 were referred from the countryside to study settings for higher level care for mothers or PTIs (Table, 1).

Gestational age of the PTIs ranged from 28–34 weeks with birth weights ranging from 1000 to 1880 grams. More than half (14 PTIs) were born via vaginal delivery. Eleven PTIs were born due to premature rupture of the membrane. The most comorbid conditions of PTIs were neonatal sepsis (n = 14). The hospitalization duration of PTIs ranged from 8 days to 41 days and the majority (n = 15) required hospitalization for 15-30 days (Table 2).

## Themes

The findings of this study are structured into three themes and 10 sub-themes (see Fig 1).

### Theme 1. Diverse experience of care and support

Parents required care and support from health personnel to cope with the situation of their PTIs hospitalization. Parents had frequent interaction and communication with the nurses. They acknowledged proper and timely communication and behavioral responses from the nurses as well as feeling emotionally supported during their NCU stay. However, about one-third of participants were dissatisfied with the communication patterns and support received

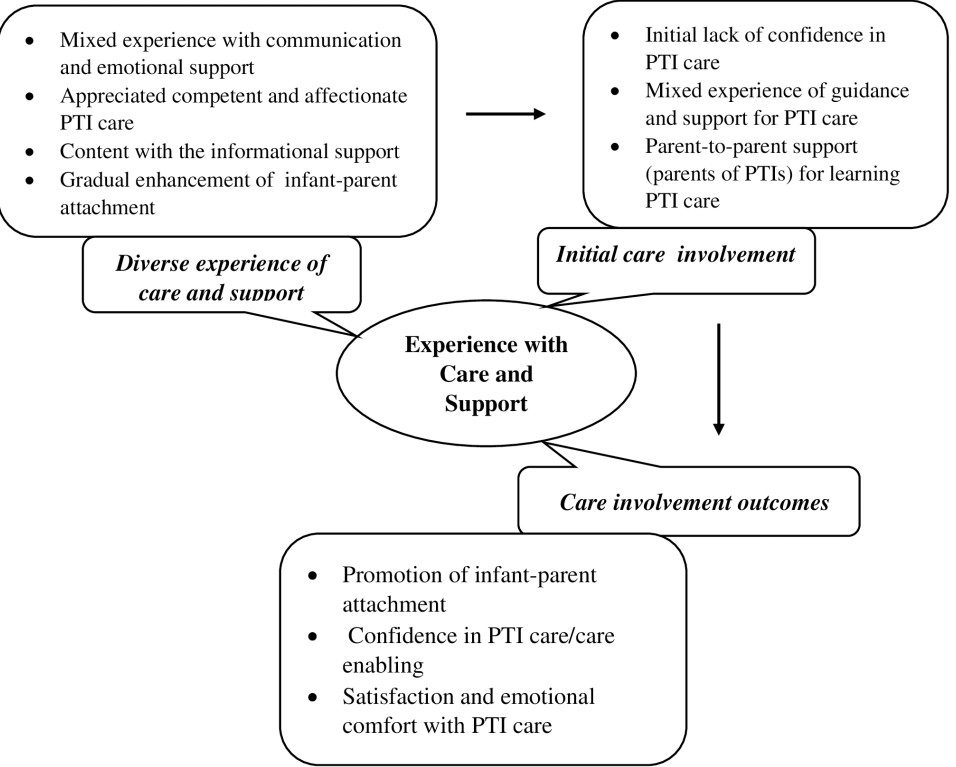

**Fig 1. Sub-themes and Themes on Parents' Experience with Preterm Infant Care and Support during Hospitalization of Their PTIs in NCUs.**

during their NCUs stay. Parents appreciated the care provided to their PTIs and contended for informational support.

**Mixed experience with communication and emotional support.** Parents experienced cordial and friendly behavior from nurses. They received responses to their questions and concerns and felt a congenial social environment on the unit, *"Nurses behaved very well with us and talked openly like a friend. I was comfortable going there for KMC and other baby care and to talk and ask them any queries."* [Mother (M)11]

They experienced nurses' behavior better than expected. A father shared,

> *"We communicated with nurses throughout the day. They were so caring, supportive, polite and friendly even in busy situations. They communicated and behaved without being irritable and annoyed. I had heard about their ridiculous behavior. But, I found, their behavior as warm and amiable."* [Father (F) F1]

Some mothers appreciated the friendly responses and genuine concern shown by the nurses, *"Nurses and staff behaved nicely, showed concern about my health condition, and reassured for my baby's condition. Sometimes, they tried to amuse me when I looked sad." (M 16,* Primi mother*)*

However, some parents expressed their disappointment regarding expectations for an ideal nurse. They shared inadequate conversation and responses from the nurses and their unfriendly behavior. Sometimes, parents felt suspicious and worried about such situations. Parents expressed,

*"We do not talk much with the nurses. We talk with them when we go there during visiting time. They talk about our baby only if we ask for some problem or doubt."* (M4)

*"We expect a polite response from nurses like the way we talk to them. But it doesn't happen like this. Some are nice but not all. Some of them didn't answer our questions. Then I felt like something bad was happening to our baby. I didn't discuss it with anyone or say a word because our baby is inside there."* (F4)

Some parents experienced rude and discouraging conversations in NCUs. They expressed a lack of a congenial environment in the unit, inadequate concern for parent-infant bonding, lack of reassurance, and genuine concern for their situation (stress related to separation and their PTI's condition) though they provided good care to their PTIs. A mother voiced,

*"We were eager to know about our baby's condition during our scheduled daily visit. But we hesitated to ask the nurse because of their rude responses sometimes. Last time, I asked about the fast breathing of my baby. A nurse said that the baby would not be in the NICU if the baby was normal."* (M 6)

Parents shared inadequate responses for their emotional distress. A parent said, *"Our baby was very sick in NICU, we were far from our family and friends. We were under great stress. Nobody was there to show any concern and reassurance to us."* (M9)

**Appreciated competent and affectionate care to PTIs.** Most of the parents appreciated the care provided for their PTIs. Some parents who observed care provided by nurses were impressed with the handling and affection shown by nurses. Others were contented with the improvement of the PTIs from critical condition. Mothers expressed gratitude for the timely feeding and for the care given to their sick PTIs in their absence.

*"I was unknown of caring for such a small baby. I saw such good care there. We can't give that much interest to others' babies. But they cared for babies so well that it looked like they were caring for their own babies. I was surprised. They were prepared for such good care."* (M3).

*"They cared for the babies very well, focusing more on the critical cases. They used to feed, cuddle the babies, keep them clean, and care with love and interaction. The way nurses showed their affection and care to the baby and mother was amazing."* (F1)

**Content with informational support.** Parents wanted timely adequate information about their critically sick PTIs separated from them. Daily and situational counseling by doctors was the key source of information about their PTIs, *"Baby's condition was informed while daily counseling by doctors. They informed us about the weight, prognosis, improvement, any problems, feeding amount, and so on."* (F3)

Similarly, mothers received information regarding baby care matters as well as unit rules and administrative requirements during their contact and interaction with the nurses during their unit visits. They expressed their satisfaction like this, *"They (nurses) explained about the unit schedule, baby's general condition, and baby care like KMC. They instructed hand washing and other precautions to enter the unit. They also explained for our queries about our babies."* (M7)

When it was time to transfer their PTIs from the NCU to mothers in mother-baby unit nurses instructed parents regarding the identification of the possible signs or symptoms of illness (danger signs) in the PTIs. They shared their gratification like this, *"They told me many*

*things that I need to be careful, e.g., for breathing pattern, warmth of the baby, stimulate the baby if bluish discoloration of the baby's face. They also instructed about Kangaroo mother care to keep warm and expressed breast milk (EBM) feeding for the baby." (M13)*

**Gradual enhancement of infant-parents attachment.** Mothers who had a strong desire to visit and be close to their PTIs experienced separation, inadequate attachment, and deprived maternal role during separation from their PTIs after birth. Parents had different experiences with visiting their PTIs in NCUs at different facilities. There was usually limited access to parents during the initial days and the visits increased considerably after starting the EBM feeding.

> *"We wished to visit our baby frequently, be close and love our baby at least for a moment during our visit but we were allowed just once a day to see from near for a while without touch, and interaction. One day, they scolded us when my husband gently touched our baby's head" (M9).*

Mothers felt anguish and loss of control for being forbidden to care for and to be close to their PTIs during the visits to NCUs, *"When I saw my baby feeling uneasy, I wished to comfort her. But I couldn't even touch her. I have to get permission from others to see my baby. I feel very helpless as a mother." (M4)*

However, one-third of parents shared their satisfaction with encouragement to visit and stimulate their PTIs even during a critical period. A father who frequently visited NCU when his spouse was unable to shared, *"Even during the initial critical condition, nurses called us, especially mother to visit our baby. They suggested touching and massaging him to promote his recovery. Though, only parents were allowed in the unit. (F1)*

Gradually with the improved condition of the PTIs and the infant starting to feed, mothers were allowed to visit their PTIs and get involved in their PTI's care, *"Later on, after starting feeding, I was called there two-hourly for feeding. At that time, we used to touch, hold, interact with him, feed breast milk and change his diaper." (M 18)*

The majority of mothers had opportunities to visit and interact with their PTIs while they went to provide EBM. However, mothers from one setting shared that they couldn't visit their PTIs during this time, *" We express breast milk in a cup in a separate room labelled with our baby's bed number and leave there for feeding. We can't visit our baby at that time and try to see inside from a glass window." (M5)*

Whereas fathers were allowed to visit their PTI once a day or when their spouse was unable, *"When she (mother) was unable, I visited there frequently. Later, she had to go there to feed frequently. So, I attend the parents' counseling or their call. I can visit daily in the evening if I like." (F2)*

## Theme 2. Initial involvement in PTI care

When PTIs were in critical condition at the NICU, they were cared for by the nurses with limited maternal/parental access or involvement in their PTI care. With progress in their PTI's condition, mothers were gradually more actively involved in providing care to their PTIs. The hands-on care like EBM feeding, handling, KMC was challenging for mothers initially. They felt inadequacy in their caring skills and felt the need for additional support and care guidance from nurses. In addition to instruction and guidance of the nurses, their skills for caring for PTIs were facilitated by the other mothers of PTIs in the NCU. This theme emerged from three subthemes (Fig 1).

**Initial lack of confidence in PTI care.** When parents were asked to be involved in their PTI's care once the PTI's condition improved, they verbalized a lack of knowledge regarding the special care required for a PTIs, *"After the improved condition of my baby, I went there*

frequently to provide EBM. They instructed me to hold and feed the baby. We had no idea about small baby care like EBM feeding, KMC, holding, and handling." (M 7)

Mothers felt difficulty and hesitation in caregiving, and fear of accidental injury and harm to their PTIs during initial care involvement, "It is uneasy to hold and handle my baby. She is so small and weak. I feel fear of injury and bit nervous to put on clothes to her." (M17)

The EBM feeding was the most difficult task for mothers. They had a fear of harm due to possible improper technique as well as being exhausted, especially at night. A mother expressed, "Cup feeding was very difficult initially. I used to be worried if something happened to her or feared of choking and spilling. It was also very time-consuming as the baby sleeps too often while feeding. (M 13)

Providing KMC in the initial days was also uncomfortable for mothers, "Initially, keeping baby in keeping baby attaching in my chest (KMC) was so uncomfortable, feeling hot, uneasy. I used to hold the baby with one hand due to fear that he may slip and ask the sisters frequently to stop it."(M18)

**Mixed experience of guidance and support for the PTI care.** Mothers had a major role in their PTI care, and fathers were also sometimes involved. Parents shared different experiences of guidance and support regarding their PTI's care. Some parents were impressed by the support they received at the NCU and appreciated the encouragement, instructions, and guidance regarding hands on care of their PTI, "In NICU, they instructed and guided me by showing how to feed, and bathe. They advised me to observe carefully and provide care the next time encouraging about care enabling after receiving the baby". (M11)

Most mothers reported feeling secure in caring for their PTI under the nurses' supervision as this provided additional support in case of a possible problem. A mother shared, "After showing me, they observed and advised the correct technique for initial feedings. They encouraged me saying that they were there if anything happened. Gradually, I felt comfortable feeding her and changing diapers during scheduled visits." (M17)

Mothers also appreciated encouragement for their PTI care involvement. A mother whose PTI was only 1,000 grams at birth and kept in NCU for 41 days shared, "In NICU nurses encouraged by saying that you will get confidence only after you care for your baby yourself. I followed them. I hadn't imagined that I could, but it became possible." (M3)

Nevertheless, about four mothers shared their perceived experience of inadequate care guidance at NCUs. Mothers shared, "I am doing it myself. They just advised me to express breast milk. They didn't teach me how to express and feed the baby. They told me to wake up my baby if she was sleepy while feeding." (M5).

> "For the first time, they kept my baby on KMC by putting the baby's bare chest on mine and then tied us with a cloth and again wrapped with another cloth on top of that. From the second day, I did whatever I grasped at that time." (M4)

A mother whose PTI was discharged from NICU with inadequate confidence in breastfeeding expressed,

> "It has been 3-4 days since I began to hold and feed my baby. They have instructed me to spoon-feed and breastfeed him alternately and continue the same after going home. I am still not confident to feed my baby yet. It would be easier if I could learn how to feed him properly before being discharged. After going home, my mother-in-law might help in feeding and other baby care." (M6)

**Parent-to-parent support for learning PTI care.** Mothers used to gather in the unit during feeding time. This situation was an opportunity for new mothers to learn PTI care

through observation or guidance by experienced mothers. A mother who had reared a normal first child stated, *"I saw mothers were feeding with a special spoon (pallade), doing KMC in the unit. Then I knew that my baby also needs those and determined to do similarly." (M1)*

Another first time mother had a medical problem and inadequate family support expressed her experience of peer support like this, *"Here (KMC unit) everybody is feeding and caring for their babies. I am also doing the same. They help and encourage and correct me with the baby care. Other mothers are helpful and support me in various ways." (M12)*

They felt that the opportunity to observe how other parents caring for their PTI at NCUs was useful in learning care for their PTI, *"Observing other mothers' doing, it was easy to follow when nurses helped KMC. Now I am feeding him, providing KMC as guided. (M16).*

## Theme 3. Care involvement outcomes

Mothers shared a positive experience with the opportunity to get involved in the care of their PTIs. Initial hospital stays with the PTI separated in NCUs were very tedious and stressful due to uncertainty of their PTI's condition and future progress as well as deprivation in maternal role. Gradually involvement in their PTI care like providing EBM became the means to visit their PTIs. Further progress of the maternal role like KMC, EBM feeding or breastfeeding were the opportunities for bonding with their PTIs, and for care enabling which led to parental satisfaction and emotional wellbeing. The theme is based on the following different subthemes (see Fig 1).

**Promotion of infant-parent attachment.**  In the situation of infant-parent separation, parenting or caring was opportunities were the means to visit and be connected with their PTIs. Mothers had feelings of closeness, love, and affection while providing care. A mother shared her feeling *"I feel immense pleasure and being a mother when keeping my baby in KMC." (M11)*

Another mother who was deprived of closeness for a long time with her PTI expressed, *"We were allowed just to see during visiting time. Nothing else but KMC and sucking physio (exercise) provided opportunities to be close to my baby." (M4)*

Another first-time mother shared her breastfeeding experience, *"During breastfeeding, I felt closer to her like having a heart-to-heart connection. Lots of mothering love flows towards my baby." (M14)*

Most of the parents were able to be involved in their PTI's care in SNCUs. Parents' happiness was related to the progress of their PTIs and also to the bonding and parenting roles. A mother expressed, *"It was quite boring when the baby was in NICU. Here (in SNCU), we visit our baby three-hourly and are involved in care. Now I am so glad and feel like being a mother." (M9)*

**Confidence in PTI care/care enabling.**  Parents, especially mothers, experienced care or parenting confidence with regular care involvement for their PTI. In particular, mothers who were involved in care of their PTI's within the unit, provided care themselves in KMC unit or mother-baby unit and habituated for care before going home. Mothers expressed,

> *"I didn't know about KMC. I did it for around 2 hours daily with their (nurses') guidance and support in the unit. On the fourth day, they shifted my baby. Now, I am doing it most of the time here in KMC unit myself." (M 18)*

> *"Baby care was very difficult initially. Now, I am comfortable as I am caring for some days myself and can provide this much care after going home." (M13)*

> *"In the beginning, I felt some kind of hesitation, and difficulty. But now I am doing all the care as instructed." (M3)*

Mothers were also able to understand their PTI's behavior cues and deal with their PTI's care needs. A mother shared,

> " *It was difficult to change his clothes and to feed him. After providing care for a week myself, it is easier now. I can understand my baby and what he likes. Sometimes, he wakes up and provides facial expression, sucks his finger and I feel he wants to feed. I can interpret his cry in different situations.* " (M11)

> "*Baby care was very difficult initially. Now, I am comfortable as I am caring for some days myself and can provide this much care after going home.*"

**Satisfaction and emotional comfort.** Mothers shared a decreased in their initial stress and worry related to separation, role deprivation after regular visit and involvement in their parenting role and PTI care. Mothers were content about their parenting roles like this, *"Now in SNCU, we visit, hold and care for our baby. I am doing KMC for about three hours, changing his diaper, and feeding him. The condition is improving. I am busy caring for him and am happier now."* (M9) Their boring time in the postnatal ward was changed to busy routines and positive engagement into their PTI's care, *"Previously, the days were long and boring. Now, time passes well with scheduled visits to the unit and baby care."* (M 16)

Mothers reported experiencing emotional comfort after involvement in the parenting role and caring for their PTI. Undoubtedly, the reason for their happiness was the improvement in outcomes of their PTI but the attachment and care involvement enhanced their positive feelings further. According to a mother, *"While caring for her, I feel close with her and being a mother. I didn't feel the pain of an operated wound."* (M14)

Mothers also reported being happy about their contribution to the recovery of their beloved PTIs. They perceived maternal care as more comforting, gentle, and affectionate to their PTIs as well and helpful to improve condition of their PTIs. A first-time mother shared, *"I think that my baby feels comfortable with KMC. It is easier to identify her condition and breathing status continuously. Today, her weight has increased by 50 grams. I feel she is comfortable with more gentle mothering care."* (M 16)

On the unit, mothers observed the hectic schedule of nurses and some of them shared their desire to support the nurses by providing care for their PTIs. So, the nurses could provide care to the critically sick PTIs or PTIs whose mothers couldn't attend the unit. A mother shared, *"Nurses have to care for so many babies. Some mothers couldn't attend to their babies due to their illness or operation. Sometimes, many babies keep crying. If I care for my baby myself, they can care for other babies.* " (M18)

## Discussion

The Findings of this study indicate that parents generally had positive experiences with care and support received at NCUs. However, some parents shared their experiences of receiving inadequate support, communication, attachment opportunity and care guidance.

Proper communication by NCU staff is the most important need for parents of PTIs in NCUs. It is important for maintaining relationship, exchanging information about PTIs and enabling for care of PTIs [23,26]. Evidence from developed countries (Netherland and UK) reported parents' higher satisfaction with the communication provided by nurses and they appreciated empathy, interpersonal interaction, emotional support, politeness, and cordial responses from nurses [26,35]. While the findings of this study indicated that parents had mixed experience with the communication patterns and emotional support received from nurses at the NCUs. Prior studies in low middle-income countries nations like Jordan, and

Ghana reported adequate and inadequate communication and emotional support in NCUs [36,45]. Similarly, parents in other studies [37–39] from different global contexts reported limited communication, interpersonal interaction, emotional support, and cordial responses from nurses. Parents also reported limited communication with nurses such as having answer(s) to their questions and sometimes even inadequate responses for their questions and concerns. Like previous study [46], parents who were far from their family (referred to from country sides) felt greater support need. In other studies [14,36] also, parents expressed the need for more empathetic approach, communication, and emotional support while their PTI are admitted to the NCUs in Nepal. Prior studies reported workload or inadequate nurse-infant ratio as the reason for such gap in communication and emotional support [46,47].

Having information about the condition and progress of their PTI regularly, and receiving accurate answers to questions are among the most important needs reported by NCU parents [28,48]. Current study revealed that parents had positive experiences of receiving information about the PTI's condition and progress. In each setting, daily parents counseling session is conducted by the doctors to inform the condition, progress and further plan of the infants to parents. Somewhere, nurses are also involved in the session. They were also encouraged to ask their questions and concerns. Parents received instruction about the unit rules, related administrative rules of the hospital, PTI care, maternal self-care from the nurses. As the nurses were available all the times in the unit, parents in present and prior study [29] found nurses as more approachable health care personnel. Previous studies also indicated parents' higher satisfaction with nurses' information sharing in NCU [42,49]. Prior evidence indicated inconsistencies across the health care team's communication, information sharing, clarifying queries, and the emotional and practical support provided to parents [46].

Parents in the present study as well as previous studies [35,42,45,50] in different contexts globally valued the competent and affectionate care of their PTIs by nurses. Parents in present study observed the provided care and appreciated the way nurses handled, cared for and showed love and affection to their PTIs. Parents expressed their sincere gratitude for recovery of their PTIs with the quality care by nurses in NCUs which is supported by other studies [29,36]. However, some parents felt inadequate care for their PTIs by nurses such as inattentiveness to their PTI's crying.

A few parents reported encouragement for contact and closeness even during critical condition of PTIs in NICU. Parents expressed unfulfillment of their attachment need with their infants while their PTIs were separated from them, especially in NICU. Finding is supported by previous studies in different contexts [14,16,41,45,51] Some parents in the present study shared about restrictions for attachment activities like touch, massage and interaction with their PTIs in NICU. Evidence indicated inadequate attachment and altered parental role as the stressors among parents in NCUs [13,14,16]. There are still some limitations in care practice like restriction in parental visit, and attachment activities considering that parents' involvement in NCUs can increase the infection rate in NCUs [17]. Similar inadequate attachment and bonding opportunity was related to the unit rule as well as mothers' health conditions [14,49]. Parents in study context respect health personnels and follow their suggestions for the favorable outcome of their infants though they have emotional difficulty [17].

With the improved condition of the PTIs, mothers were gradually provided opportunities to interact with their PTIs and engage in their parenting roles gradually, such as EBM feeding, holding. and KMC in NCUs. The initial involvement in care of their PTIs strengthened their parental feeling and it was challenging as well [14,16,17]. Due to early and prolong separation and special care need of PTIs, new parenting roles can be unfamiliar, difficult, and different for parents irrespective of their parity and socio-cultural background. Previous studies in various context also reported similar findings [14,16,27,42]. Parents felt inadequate skills and

confidence manifested by hesitation to handle and care, fear of injury and harm and uneasiness for providing care to their PTI [29,52].

Parents' involvement in activities like touching, holding, and interacting with their PTIs were reported as being meaningful to parents [15,16,50]. Parents feel more comfortable, and empowered when providing hands-on care to their PTIs [23,27]. The involvement in the care of their PTIs like EBM feeding, KMC was new for mothers even having previous children. Therefore, mothers irrespective of age group, parity, educational status expressed need for guidance, enabling environment to provide hands-on care to their PTIs in the initial days of their care involvement [23,27,30]. Parents expressed satisfaction with care involvement after observing the nurses and being provided an opportunity in a supportive manner [27,45,50].

Some mothers in this study shared inadequate guidance for hands-on care, and lacking in parenting confidence before discharge resulting in stress and worry for care of their PTIs [14,15,29,42,45]. Prior studies reported fathers' feeling of excluded in the care of their PTIs in NCUs [53,54]. Fathers in this study also expressed concern regarding their involvement in PTI care.

In addition to support for parents provided by nurses, peer-to-peer support is effective for initial care learning [18]. In addition to reassuring one another, experienced mothers were a source of inspiration and encouragement for mothers with less confidence for hands on care (parenting role) [15,18,45]. Peer-to-peer support is identified as the most facilitating and supportive aspect of developing one's maternal role in NCUs [18,45,55–57] Thereby resulting preparedness for discharge and transitional care for PTI. Parents indicated creation of supportive environment for parenting roles with rooming in with other parents having PTIs [14,45].

Parents shared that some days of engagement in hands-on care enhanced their confidence in PTI care [15,16,52,58,59]. Consistent to earlier findings, parents also felt enhanced attachment with their PTIs, emotional comfort and satisfaction [15,58,60]. Parents also felt a sense of contribution and accomplishment.

Although this is the first qualitative study examining parents' perspectives of NCUs and PTIs care in Nepal, this study has certain limitations. The study focused on parents of the PTIs with positive prognosis who are transitioning to low acuity care or close to being discharged from NCUs. Therefore, findings can only be generalized to similar populations. Considering the greater maternal roles, more mothers were included in the study. However, during the initial critical phase, fathers used to be the main person visiting their PTIs in NCUs considering the medical condition after delivery, the requirement of rest, and emotional wellbeing of postpartum mothers. Some mothers with complicated conditions like antepartum hemorrhage couldn't attend their PTIs initially. For broader exploration of the care and support status in NCUs both parents were recruited in the study. However, only five fathers were included in the study considering it as the minimum sample size for in-depth interviews and exploration of substantial information for the study. The inclusion of both parents in equal ratio and study among fathers might be another areas for exploration. Although more than half of the participants were referred from rural areas, the study was conducted at tertiary hospitals of Kathmandu Nepal. Some key information about their NCUs experience could have been missed based on the depth of information participants were willing to share. Some meanings inherent to participants' expressions might have been lost during analysis, despite the rigorous checks performed by all authors, including measures like member checking and rigorous language translation.

## Conclusion

Parents' experience with care and support in the NCU involves the recipience of competent and compassionate care for their PTIs, contentment with informational support, and gradual

enhancement of attachment activities with their PTIs. Parents, specifically mothers, engage in hands-on care for their PTIs progressively, generally in SNCU.

However, some parents have experience of receiving inadequate communication, emotional support, cordial responses and support for involvement in care of their PTIs in NCUs. The initial involvement in PTI care is challenging for parents requiring care guidance and support. In addition to nurses' support, support by peers also known as peer-to-peer support (support by parents of PTIs admitted to NCUs) facilitate parenting role development among parents. Involvement in hands on care of PTI in NCU is worthwhile for promotion of infant-parent attachment, development of PTI care confidence, and enhancement of emotional wellbeing of the parents. These are essential components for parenting confidence in caring infants and better hospital to home transition of PTI admitted in NCUs.

Findings from this study represented a deeper understanding of parents' experiences concerning care and support received in NCUs in Nepal, with a specific focus on the support provided by the nurses. However, it is crucial to acknowledge that the nurses' practice can't be completely separated from the clinical situation. The provision of care and support in the NCU is a collaborative effort involving various professionals, with nurses playing a pivotal role. Thus, future research should explore parents' experience at NCUs with multidisciplinary healthcare team members including physician and other healthcare teams.

## Implications in clinical practice

Noteworthy among the findings from this study is that parents appreciated the care provided to their PTIs and anticipated more parental support in NCUs in the form of effective communication, opportunities for infant attachment and guidance and support for involvement in the care of their PTIs. Their positive experience regarding involvement in the care of their PTIs in NCUs indicates the need for enhanced practice in NCUs. Parents participation in PTI care is effective with adequate guidance, encouragement and in parent-friendly NCU environments. Therefore, findings from this study provide incredible feedback for nurses and other clinicians for enhancing their practice. Parent to parent (peer to peer) support during initial care learning is another important aspect for clinical application. In the situation of inadequate nursing personnel in NCUs, mobilization of peer-to-peer support system by experienced NCU parents under supervision might be valuable

The varied experience of care and support among parents across different NCUs settings in Nepal denoted distinct variation in practice. The evidence highlights the need for improved NCUs care with universal policies and strategies by the government for parental support and a parent-friendly NCU environment. Endorsement and enhancement of family centered care in NCUs would be remarkable. However, it requires the development of guidelines, enhancing knowledge and positive mindset among neonatal nurses as well as providers through proper training, education, and provision of adequate resources including human resources. Training for nurses and other providers/clinicians on parental support at NCUs and establishing a parent-friendly NCU is valuable.

Though the findings of this study were derived from parents' perspectives of care and support experiences from nurses, it represents the actual circumstances of the NCUs in Nepal and other resource limited countries. Thus, additional research and evidence-based interventions at NCUs settings are required.

## Supporting information

**S1 Data. Data analysis matrix, care and support.**
(DOCX)

## Acknowledgments

The authors express their gratitude to study settings, participants of the study, language editor Associate Professor Dr. Mary Wunnenberg, at Rutgers University School of Nursing – Camden, USA and and all the people who directly or indirectly contributed into the study.

This manuscript is part of the exploratory sequential mixed method study of the PhD dissertation.

## Author contributions

**Conceptualization:** Tumla Shrestha, Archana Pandey Bista, Sarala Joshi Pradhan, Sangita Pudasainee-Kapri, Madhusudan Subedi.

**Data curation:** Tumla Shrestha.

**Formal analysis:** Tumla Shrestha.

**Investigation:** Tumla Shrestha.

**Methodology:** Tumla Shrestha, Archana Pandey Bista, Madhusudan Subedi.

**Project administration:** Tumla Shrestha, Archana Pandey Bista.

**Resources:** Tumla Shrestha, Archana Pandey Bista.

**Supervision:** Archana Pandey Bista, Sarala Joshi Pradhan, Madhusudan Subedi.

**Validation:** Tumla Shrestha, Archana Pandey Bista, Sarala Joshi Pradhan, Sangita Pudasainee-Kapri, Madhusudan Subedi.

**Writing – original draft:** Tumla Shrestha, Sangita Pudasainee-Kapri.

**Writing – review & editing:** Tumla Shrestha, Archana Pandey Bista, Sarala Joshi Pradhan, Sangita Pudasainee-Kapri, Madhusudan Subedi.

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
