## [Decision Letter · Decision Letter 0]

17 Dec 2024

PONE-D-24-41221Unveiling parents’ lived experience with preterm infant care and support in neonatal care units of public hospitals in Nepal: A phenomenological inquiryPLOS ONE

Dear Dr. Shrestha,

Thank you for submitting your manuscript to PLOS ONE. After careful consideration, we feel that it has merit but does not fully meet PLOS ONE’s publication criteria as it currently stands. Therefore, we invite you to submit a revised version of the manuscript that addresses the points raised during the review process.

We look forward to receiving your revised manuscript.

Kind regards,

Alemayehu Getahun Kumela, Ph.D.

Academic Editor

PLOS ONE

Journal Requirements: When submitting your revision, we need you to address these additional requirements. 1. Please ensure that your manuscript meets PLOS ONE's style requirements, including those for file naming. The PLOS ONE style templates can be found at https://journals.plos.org/plosone/s/file?id=wjVg/PLOSOne_formatting_sample_main_body.pdf and https://journals.plos.org/plosone/s/file?id=ba62/PLOSOne_formatting_sample_title_authors_affiliations.pdf 2. Please include a complete copy of PLOS’ questionnaire on inclusivity in global research in your revised manuscript. Our policy for research in this area aims to improve transparency in the reporting of research performed outside of researchers’ own country or community. The policy applies to researchers who have travelled to a different country to conduct research, research with Indigenous populations or their lands, and research on cultural artefacts. The questionnaire can also be requested at the journal’s discretion for any other submissions, even if these conditions are not met.  Please find more information on the policy and a link to download a blank copy of the questionnaire here: https://journals.plos.org/plosone/s/best-practices-in-research-reporting. Please upload a completed version of your questionnaire as Supporting Information when you resubmit your manuscript. 3. In the online submission form, you indicated that "Most of the relevant data are within the manuscript. Further information if required will be provided by the first author." All PLOS journals now require all data underlying the findings described in their manuscript to be freely available to other researchers, either 1. In a public repository, 2. Within the manuscript itself, or 3. Uploaded as supplementary information.This policy applies to all data except where public deposition would breach compliance with the protocol approved by your research ethics board. If your data cannot be made publicly available for ethical or legal reasons (e.g., public availability would compromise patient privacy), please explain your reasons on resubmission and your exemption request will be escalated for approval.

Reviewers' comments:

Reviewer's Responses to Questions

**Comments to the Author**

1. Is the manuscript technically sound, and do the data support the conclusions?

Reviewer #1: Yes

Reviewer #2: Yes

2. Has the statistical analysis been performed appropriately and rigorously? 

Reviewer #1: Yes

Reviewer #2: Yes

3. Have the authors made all data underlying the findings in their manuscript fully available?

Reviewer #1: Yes

Reviewer #2: No

4. Is the manuscript presented in an intelligible fashion and written in standard English?

Reviewer #1: Yes

Reviewer #2: Yes

5. Review Comments to the Author

Reviewer #1: Dear authors

Thank you for your efforts to do this research. I have provided some comments and suggestions below to help enhance the clarity and impact of your study.

Abstract:

Please include the duration of the study.

Introduction section:

I suggest that the authors enhance the depth and richness of the introduction section by referencing similar studies. For example, I recommend the following article:

https://journals.sagepub.com/doi/full/10.1177/17455057221104674

Method section:

Please merge the "Research Instrument" section with "Data Collection."

Lincoln and Guba, in their latest edition, have added the criterion of "authenticity" to the four existing criteria for data rigor. Please include this criterion as well, along with an explanation of how it has been implemented.

Please explain the maximum variation in sampling.

Results section:

Regarding the findings, I am fully aware that the authors have deeply immersed themselves in the data and have a better understanding than anyone else. However, as a reviewer, I would like to offer some suggestions on certain themes and sub-themes from a reader’s perspective to enhance clarity and understanding.

Please ensure that the abbreviation "EBM" is written out in full upon its first use. Although it was mentioned in the abbreviations section at the end, it was not expanded in the main text when it first appeared.

The theme "Support and Care" seems to need a more fitting concept. After reading its content and sub-themes, I realized that it doesn’t merely depict "support and care" but rather presents a continuum that includes both desirable and undesirable support and care. Since a theme, as an abstract concept, should encompass all sub-themes, it would be better to consider a title that aligns more appropriately with this.

I believe the sub-theme "Gradual enhancement of infant-parent attachment" can be considered as an outcome and overlaps with the sub-theme "Promotion of infant-parent attachment". If this is not the case, the authors should clarify.

Discussion Section:

I suggest that the authors enhance the depth and richness of the discussion section by referencing similar studies. For example, I recommend the following articles:

https://onlinelibrary.wiley.com/doi/full/10.1155/2021/6697659

https://onlinelibrary.wiley.com/doi/abs/10.1111/nicc.13110

Regards

Reviewer #2: Dear authors

I would like to thank you for giving me the opportunity to review the manuscript entitled “Unveiling parents’ lived experience with preterm infant care and support in neonatal care units of public hospitals in Nepal: A phenomenological inquiry”. The manuscript presents an interesting and potentially valuable study. However, it should be carefully revised. Please find my comments:

- While the themes are comprehensive, their presentation could be made more concise, with a clearer distinction between overlapping subthemes (e.g., communication and emotional support).

- Consider using a visual framework to summarize the relationships between themes.

- Expand on how cultural factors specific to Nepal influence parents’ experiences and expectations in neonatal care units.

- Compare findings more explicitly with existing literature from similar LMICs to provide a broader context.

- The sample is predominantly mothers, with only a few fathers included. Address this limitation more explicitly in the discussion and suggest strategies for future studies to achieve gender balance.

- Consider exploring the implications of variations in parental education and socioeconomic status on their experiences.

- Provide more actionable recommendations for healthcare providers and policymakers. For instance, suggest specific training programs for nurses to enhance communication and emotional support.

- Highlight how peer-to-peer support systems could be formally integrated into neonatal care.

- Ensure consistency in terminology (e.g., preterm infants (PTIs) versus preterm babies).

- Revise some repetitive phrases (e.g., emotional support and care guidance) to improve readability.

- Correct minor typographical errors in the text.

6. PLOS authors have the option to publish the peer review history of their article (what does this mean? ). If published, this will include your full peer review and any attached files.

**Do you want your identity to be public for this peer review?** For information about this choice, including consent withdrawal, please see our Privacy Policy .

Reviewer #1: No

Reviewer #2: **Yes: ** Confirm

---

## [Author Response · Author response to Decision Letter 1]

20 Jan 2025

Dear Editor and Reviewers,

Thank you very much for providing expert suggestions to revise this manuscript and clarify further. We have included a detailed description below along with the updated manuscript accordingly. Please let us know if you have any questions or need additional information.

Based on the suggestions provided, this manuscript is

- copy edited by a professional native speaker,

- revised following the PLOS ONE manuscript style and

- extensively reviewed by the authors according to suggestions provided by the reviewers and academic editor.

A completed version of PLOS’ questionnaire on inclusivity in global research is uploaded as supplementary information.

Regarding the data underlying findings, most of the data that supports findings of this study is included in the manuscript itself and remaining data are attached as supplementary files: “Data Analysis Matrix, Care and Support”.

Thank you!

---

## [Decision Letter · Decision Letter 1]

27 Jan 2025

Unveiling parents’ lived experience with preterm infant care and support in neonatal care units of public hospitals in Nepal: A phenomenological inquiry

PONE-D-24-41221R1

Dear Ms Tumla Shrestha,

We’re pleased to inform you that your manuscript has been judged scientifically suitable for publication and will be formally accepted for publication once it meets all outstanding technical requirements.

Kind regards,

Alemayehu Getahun Kumela, Ph.D.

Academic Editor

PLOS ONE

Reviewers' comments:

Reviewer's Responses to Questions

**Comments to the Author**

1. If the authors have adequately addressed your comments raised in a previous round of review and you feel that this manuscript is now acceptable for publication, you may indicate that here to bypass the “Comments to the Author” section, enter your conflict of interest statement in the “Confidential to Editor” section, and submit your "Accept" recommendation.

Reviewer #1: All comments have been addressed

Reviewer #2: All comments have been addressed

2. Is the manuscript technically sound, and do the data support the conclusions?

Reviewer #1: Yes

Reviewer #2: Yes

3. Has the statistical analysis been performed appropriately and rigorously? 

Reviewer #1: N/A

Reviewer #2: N/A

4. Have the authors made all data underlying the findings in their manuscript fully available?

Reviewer #1: Yes

Reviewer #2: (No Response)

5. Is the manuscript presented in an intelligible fashion and written in standard English?

Reviewer #1: Yes

Reviewer #2: Yes

6. Review Comments to the Author

Reviewer #1: Dear authors

Thank you for addressing my comments. I think there is a good improvement in the manuscript. I have no additional comments.

Best regard

Reviewer #2: (No Response)

7. PLOS authors have the option to publish the peer review history of their article (what does this mean? ). If published, this will include your full peer review and any attached files.

**Do you want your identity to be public for this peer review?** For information about this choice, including consent withdrawal, please see our Privacy Policy .

Reviewer #1: No

Reviewer #2: No

---

## [Editor Report · Acceptance letter]

PONE-D-24-41221R1

PLOS ONE

Dear Dr. Shrestha,

I'm pleased to inform you that your manuscript has been deemed suitable for publication in PLOS ONE. Congratulations! Your manuscript is now being handed over to our production team.

Kind regards,

on behalf of

Dr. Alemayehu Getahun Kumela

Academic Editor

PLOS ONE